# Tryptophan: Its Metabolism along the Kynurenine, Serotonin, and Indole Pathway in Malignant Melanoma

**DOI:** 10.3390/ijms23169160

**Published:** 2022-08-15

**Authors:** Beáta Hubková, Marcela Valko-Rokytovská, Beáta Čižmárová, Marianna Zábavníková, Mária Mareková, Anna Birková

**Affiliations:** 1Department of Medical and Clinical Biochemistry, Faculty of Medicine, Pavol Jozef Šafárik University in Košice, Tr. SNP 1, 040 11 Košice, Slovakia; 2Department of Chemistry, Biochemistry and Biophysics, University of Veterinary Medicine and Pharmacy in Košice, Komenského 73, 041 81 Košice, Slovakia; 3KOREKTCHIR s.r.o., Zborovská 7, 040 01 Košice, Slovakia

**Keywords:** tryptophan, malignant melanoma, kynurenine, kynurenic acid, 5-hydroxyindole-3-acetic acid, 5,6-dihydroxyindole-2-carboxylic acid, indoxyl sulfate

## Abstract

(1) Background: Tryptophan metabolism is known to be one of the important mechanisms used by cancer to evade immune surveillance. Altered tryptophan metabolism was studied in patients with pigmented malignant melanoma confirmed histologically by the anatomic stage grouping for cutaneous melanoma using clinical staging on the basis of the Breslow thickness of the melanoma, the degree of spread to regional lymph nodes, and by the presence of distant metastasis. (2) Methods: Urinary tryptophan metabolites were detected by RP-HPLC method. (3) Results: In the present work, we provided evidence of altered metabolism of all tryptophan pathways in melanoma patients. (4) Conclusions: Knowledge of the shifted serotonin pathway toward DHICA formation and kynurenine pathway shifted toward NAD^+^ production could serve in the early detection of the disease and the initiation of early treatment of malignant melanoma.

## 1. Introduction

Cutaneous melanoma is one of the most aggressive forms of skin cancer and one of the leading causes of cancer death due its metastatic potential. The incidence of melanoma is increasing worldwide, and it represents 3% of all skin cancers but 65% of skin cancer deaths [1]. Melanoma is currently the fifth in men and sixth most common solid malignancy diagnosed in women [2].

Tryptophan (Trp) metabolism is associated with a variety of biological processes, including nerve conduction, inflammation, and the immune response. As an essential amino acid, it is mostly taken up by nutrition and is metabolized via three pathways: the kynurenine, serotonin, and indole pathways in mammalian cells (Figure 1). In addition to tyrosine, tryptophan and its metabolites are also involved in melanogenesis [3,4,5,6].

The primary route for tryptophan catabolism in the liver is its degradation via the kynurenine pathway (KP). It is estimated that up to 95% of dietary Trp is metabolized via the KP, of which 90% is catabolized in the liver, but the minor extrahepatic KP plays a more active role during immune activation. The enzymes *indoleamine-2,3-dioxygenase* (IDO, both IDO1 and IDO2) and *tryptophan-2,3-dioxygenase* (TDO) catalyze the first and at the same time the rate-limiting step of the pathway [7]. While TDO is mainly expressed in the liver and has a higher substrate specificity, IDO is expressed in various organs, mostly in the cells of the immune system and brain, but exerts lower substrate specificity. IDO is highly induced by the pro-inflammatory cytokine interferon γ (IFN-γ). Both enzymes give rise to N-formyl-L-kynurenine, which is in turn metabolized to L-kynurenine (Kyn) by *formamidase*. In addition to Kyn, activation of the kynurenine pathway leads to a synthesis of diverse neuroactive and immunomodulatory substances known as kynurenine pathway metabolites. Under physiological conditions, Kyn is preferentially converted into 3-hydroxykynurenine (3-HK), 3-hydroxyanthranilic acid (3-HAA), and quinolinic acid (QUIN) followed by nicotinamide adenine dinucleotide (NAD^+^) biosynthesis, an important cellular energy source. Branches in this pathway produce anthranilic acid (AA), xanthurenic acid (XA), and picolinic acid (PIC). The remaining Kyn is converted to a neuroprotective kynurenic acid (KYNA) [8,9]. Changes in the kynurenine pathway are implicated in a variety of neuropsychiatric and neurodegenerative diseases, immunological disorders, and many other disease states. Recent studies have shown that the kynurenine pathway may be involved in acute and chronic skin damage by UV radiation [8,10,11]. IDO, TDO, and L-kynurenine have been shown to accelerate tumorigenesis, proliferation, invasion, and metastasis. Dysregulation of the kynurenine pathway is thought to be a mechanism of tumor immune escape through the enzymatic activity of IDO/TDO and Kyn production. The tumor-promoting effect of Trp metabolism may be due to its effective control of the immunosuppressive microenvironment in which dendritic cells expressing IDO/TDO mediate T cell anergy [12]. Enzyme activities along the kynurenine pathway are the focus of research as they are ideal targets for inhibitors that may be valuable in the treatment of various diseases. In general, enzyme activity can be defined by the concentrations of direct derivatives of metabolites. Similarly, the Kyn/Trp ratio is used as an index of IDO activity [13,14]. Altered IDO activity is found in a wide range of human malignancies, such as breast cancer, endometrial cancer, gastric cancer, glioblastoma, lung cancer, head and neck cancer, and pancreatic cancer, hematological malignancies as well as melanoma [7,15,16]. TDO, which catalyzes the same response as IDO, is also expressed in several neoplasms, particularly in malignant glioma, hepatocellular carcinoma, bladder cancer, and melanoma [17]. The study by Asp et al. found that genes encoding kynurenine pathway enzymes, namely TDO, IDO1, *kynurenine 3-monooxygenase* (KMO), *kynureninase* (KYNU), *3-hydroxyanthranilic acid oxygenase* (HAAO), *quinolinic acid phosphoribosyltransferase* (QPRT), and *kynurenine aminotransferases* (KATs) were expressed in human skin-derived fibroblasts [11].

Tryptophan is also a precursor of serotonin (5-hydroxytryptamine, 5-HT), which is involved in the physiological regulation of several behavioral and neuroendocrine functions. Approximately 10% to 15% of exogenous Trp, which does not bind to albumin, can cross the blood/brain barrier. Although serotonin is well-known as a brain neurotransmitter, the serotonin pathway (SP) is not exclusively located in neurons originated in the raphe nuclei in the brain. Conversely, even up to 90% of the body’s serotonin production is made in the gastrointestinal tract (GIT), by the enteric nervous system and the mucosal enterochromaffin-like cells. Serotonin as a neurotransmitter is well-known in several psychiatric and neurological disorders. Increased IDO1 activity contributes to mood reduction when tryptophan availability decreases during periods of immune activation. Furthermore, serotonin is also accepted as a substrate by IDO1, mostly under conditions that cause prolonged IDO1 activity [18]. The limitation of the serotonin determination is its quick metabolization by mitochondrial *monoamine oxidase* (MAO) to 5-hydroxyindole-3-acetic acid (5-HIAA). Significant reduction in 5-HIAA levels are shown in patients with depression [19]. According to the latest findings, the serotonergic pathway is also implicated in tumor angiogenesis [6,20,21]. Several studies have demonstrated the role of serotonin and 5-HT receptor subtypes in cell proliferation, angiogenesis, invasion, migration, and metastasis. Genetic models of several cancer cells, such as lung cancer cells and melanoma cells, have shown that serotonin levels in tumors have played a key role in tumor growth [22,23]. Mitochondrial 5-HIAA has been shown to have increased production in human epidermal keratinocytes and melanoma cells [19]. In our previous study, we pointed to increased urinary excretion of 5-HIAA in melanoma patients [4]. The study of Vogliardi et al. pointed to a direct involvement of the serotonin pathway in melanin synthesis. DHICA, which is the product of tyrosine metabolization during melanin synthesis can be a noncanonical product of L-5-hydroxytryptophan in a tyrosinase-catalyzed reaction [6].

The microbial degradation of tryptophan produces a myriad of active indole-derivatives by a pathway known as the indole pyruvate pathway. Trp metabolites are thus classified together with short-chain fatty-acids and secondary bile acids among metabolites with an important role in intestinal physiology. Among the many microbial intestinal catabolites of Trp, the following have been identified and characterized: indole, tryptamine, skatole, indole-3-pyruvate, indole-3-lactate, indole-3-acrylate, indole-3-propionate, indole-3-acetamide, indole-3-acetate, indole-3-ethanol, indole-3-aldehyde, and indole-3-acetaldehyde, indole-3-acetic acid, indoxyl, indoxyl sulfate (IS), indoxyl acetate [24]. The microbiome is a key component of the tumor microenvironment and influences the initiation, support and response to cancer treatment; therefore, it is obvious that the metabolism of tryptophan through the microbial transformation to indole compounds is altered in carcinogenesis [25]. IS, the gut derived pro-inflammatory uremic toxin, together with most of the indole-based compounds, has the aryl hydrocarbon receptor (AhR, dioxin receptor) as its principal molecular target. AhR is an aromatic hydrocarbons ligand-inducible cytosolic transcription factor that regulates gene expression for immunity, stem cell maintenance, and cellular differentiation. It has long been thought that only exogenous molecules act as ligands, but recent findings confirm a huge number of endogenous ligands for this receptor. These include, not only aryl compounds synthesized by the gut microbiome, but also microbial metabolites in the skin, photoreactive Trp catabolites formed in the skin by UV radiation [24,26].

## 2. Results

The staging of the melanoma skin cancer is based on the American Joint Committee on Cancer (AJCC) tumor-nodes-metastasis (TNM) system [27]. The earliest stage, when the cancer is confined to the epidermis with no spread to the lymph nodes or to distant parts of the body, the in situ melanoma, is labelled as stage 0. The other stages are graded and numbered I to IV based on their spread, with stage IV having the highest spread. Within a stage, a classification is also made and is marked A, B, etc., which points to an increasing stage. The composition of the respondents based on the melanoma skin cancer staging is summarized in Table 1.

The urinary concentration of tryptophan and its metabolites IS, KYNA, Kyn, 5-HIAA, and DHICA was analyzed both in the melanoma patients group as well in the healthy control group and is summarized in Table 2. Data are reported as concentrations relative to actual creatinine concentration in the sample, as average ± standard deviation and as median and interquartile range. Since most of the metabolites showed a log normal distribution of values, the nonparametric Mann–Whitney U test (Wilcoxon rank-sum test) of statistical significance was used to compare the results among the melanoma patients and the healthy control group. Levels of Trp and its selected metabolites were significantly higher in the melanoma patients compared to the healthy control group (*p* < 0.001 ***). Among the studied metabolites, the highest variance in concentrations were recorded for KYNA in both groups (median 699.39 µmol/mmol of creatinine and 202.01 µmol/mmol of creatinine; IQR 1100.92 µmol/mmol of creatinine and 191.84 µmol/mmol of creatinine in melanoma patients and in healthy control group, respectively). Another important metabolite, but with a significantly lower concentration, was IS (median 76.31 µmol/mmol of creatinine, IQR 154.27 and 5.00 µmol/mmol of creatinine, IQR 6.58 in melanoma patients and in healthy control group, respectively). In the melanoma patients group, DHICA was measured below the level of detectable concentration in only one patient (median value 0.44 µmol/mmol of creatinine, IQR 15.91), but in the control group in up to 27 people, and even in those subjects, where quantification of DHICA was performed, extremely low concentrations were detected (median value 0.00 µmol/mmol of creatinine, IQR 0.01).

To understand the regulation of biochemical processes, it is helpful to look at the ratios of metabolites in the given pathways. For example, the Kyn/Trp is a helpful tool characterizing the activity of enzymes involved in the conversion of Trp to Kyn.

### 2.1. Kynurenine Pathway

The ratios Kyn/Trp, KYNA/Trp, and KYNA/Kyn associated with the kynurenine pathway are presented in Table 3. The Kyn/Trp and KYNA/Trp ratios were significantly higher, while the KYNA/Kyn ratio was not significantly different but was lower in the malignant melanoma patients compared to the healthy control (*p* values 0.004 **, 0.042 *, 0.071 ^ns^ for Kyn/Trp, KYNA/Trp, and KYNA/Kyn, respectively). The data indicate increased Trp metabolism by TDO, IDO1, and IDO2 enzymes both in the direction of NAD^+^ synthesis and in the direction of KYNA branching in melanoma patients. However, the higher KYNA/Kyn ratio in healthy individuals indicates a significant diversion of the kynurenine pathway under physiological conditions toward neuroprotective KYNA formation.

### 2.2. Serotonin Pathway

The ratios 5-HIAA/Trp and DHICA/Trp indicating Trp conversion in the serotonin pathway are presented in Table 4. A lower 5-HIAA/Trp ratio was found in malignant melanoma patients compared to the healthy control (*p* = 0.013 *). In contrast, a higher DHICA/Trp ratio was found in melanoma patients compared to the healthy control (*p* < 0.001 ***).

### 2.3. Indole Pathway

IS/Trp describing the serotonin pathway is listed in Table 4. Although the IS concentration was significantly increased in the melanoma patients group (Table 2), the IS/Trp ratio indicates the importance of the indole pathway under physiological conditions, as this ratio was higher in the healthy control group compared to melanoma patients, but the difference was not significant (*p* = 0.588).

By summarizing the results of the obtained ratios, it is possible to describe the difference in the mentioned pathways among melanoma patients and the healthy control group (Figure 2). While the concentration of all monitored parameters was higher in melanoma patients, the monitoring of the ratios showed a significant deviation in the kynurenine pathway in the direction of neuroprotective KYNA production under physiological conditions as well as a reduction in the serotonin pathway in patients with melanoma, but with a possible conversion to melanin intermediate, DHICA.

The correlation analysis revealed a statistically significant relation between the 5-HIAA, DHICA, IS, and the melanoma stage. The mentioned tryptophan metabolites correlated negatively with the melanoma stage, which sounds logical in the case of 5-HIAA and IS since these metabolites were found in patients with melanoma at a lower concentration than in healthy individuals when compared to the tryptophan level. The negative correlation between DHICA and the melanoma stage suggests that as the metabolite is present only in melanoma patients and not in healthy individuals, its presence may be among the first signs of the developing melanoma since its concentration is at higher melanoma stages again negligible (Table 5). This assumption is also strengthened by the results of the correlation analysis between the ratio of DHICA to Trp as well as the ratio of 5-HIAA to DHICA (Table 6). While the correlation between DHICA/Trp and melanoma stage is negative, the opposite is true for the correlation between 5-HIAA/DHICA and melanoma stage. In the case of these ratios, a statistically significant correlation was demonstrated not only between the melanoma stages, but also in Breslow thickness, negative for ratio DHICA/Trp and positive for ratio 5-HIAA/Trp.

One of the adverse prognostic findings of melanoma is ulceration. In Figure 3, Figure 4 and Figure 5, comparisons of tryptophan metabolites are provided based on the presence of ulceration. These comparisons did not show a statistically significant difference between the data, but highlight the insights gained from the statistical analysis. We assume that the concentration of those metabolites, the level of which was related to the actual tryptophan concentration, was higher in melanoma patients compared to healthy individuals (Kyn, KA, DHICA), with the presence of ulceration, this difference deepens. We observed this for kynurenine and kynurenic acid (Figure 3). However, suppose this mentioned difference in the presence of ulceration is not proven. In that case, it can be assumed that an increase of the given metabolites concentration is typical for the early stages of the disease, which we believe in the case of the 5,6-dihydroxyindole-2-carboxylic acid (Figure 4). This suggestion strengthens the observation of urinary DHICA concentration depending on the melanoma stage. As shown in Figure 6, the highest values are recorded in patients with melanoma stages IA and IB, while the urinary DHICA concentration decreases with increasing stages.

## 3. Discussion

Tryptophan, an essential amino acid, is a source of bioactive compounds participating in diverse physiological and pathological processes. Most dietary Trp is metabolized via three pathways: the kynurenine pathway, the serotonin pathway, and the indole pathway, with the kynurenine pathway as the most striking. Tryptophan metabolism is known as one of the important mechanisms exploited by cancer to evade immune surveillance. The cancer-associated immunosuppression has long been explained solely by the enzymatic activity of IDO. Recent evidence suggests that the products of all tryptophan metabolism pathways, especially those that act as AhR activators or AhR ligands with consequent effect as transcription factors, have an equally significant, if not even more significant, effect on tumor progression. In the present work, we provided evidence of altered metabolism of all tryptophan pathways in melanoma patients.

Local Trp deprivation caused by IDO activity causes general control nonderepressible 2-mediated (GCN2-mediated) comprehensive stress response, T cell inhibition, and decreased inflammatory response enabling cancer cells to evade immune surveillance [28]. Increased IDO expression and the decreased serum tryptophan concentration are therefore considered markers of many types of cancers and predictive markers for the poor prognoses in malignant melanoma patients [29]. Melanoma patients in our study showed significantly higher levels of tryptophan in the urine, which may not be inconsistent with the low serum tryptophan concentration in malignant melanoma patients with poor prognosis. The low level of tryptophan in the serum of patients with melanoma is explained precisely by its accelerated metabolism and degradation, which results in an increased concentration of degradation products both in serum and in urine as well as the appearance of tryptophan itself in the urine [29]. This finding is in accordance with the study of Costa et al. [30]. In addition, in our cohort, increased concentrations of monitored kynurenine metabolites compared to the original tryptophan in patients with melanoma were observed. Kynurenine is not a direct product of IDO activity, but its concentration logically increases with increased IDO activity. The Kyn/Trp ratio can evaluate IDO activity, which showed potential even as a predictable biomarker of frailty syndrome in the elderly [31]. Kynurenine levels are known to increase with age as well as in inflammatory diseases. Kynurenine was also used as a biomarker to predict an increased risk of mortality in SARS-CoV-2 infected people since it can point to an inefficient immune response accompanied by an overwhelming inflammatory reaction [32]. Meireson et al. examined the clinical significance of IDO1 expression in the serum of patients with stage I–III melanoma without prior systemic treatment [14]. High IDO1 expression in peripheral monocytes and low interferon-γ (IFNγ)-induced upregulation of IDO1 correlated with a worse outcome independent of disease stage. By comparing the Kyn/Trp ratio in our cohort, a statistically significant increase in IDO activity in patients with melanoma was confirmed. Kynurenine acts as an AhR activator that stimulates IDO expression, contributing to further kynurenine production, thus, creating a positive feedback loop. In addition, tryptophan depletion and Kyn accumulation induce the formation of immunosuppressive regulatory T cells (Tregs) that promote tumor growth. Other downstream metabolites of kynurenine, such as 3-HAA, also have an activating function on IDO expression. Comparison of the individual monitored metabolites with each other showed a higher ratio of Kyn to KYNA in patients and a higher ratio of KYNA to Kyn in healthy individuals. The formation of the immunomodulatory and neuroprotective KYNA is a branch of the kynurenine pathway from the metabolization in the direction of NAD^+^ formation, during which explicitly toxic intermediates are also produced, e.g., anthranilic acid and quinolinic acid. The increased energy requirements of the cancer cell also explain the significant diversion toward NAD^+^ production in melanoma patients. In this regard, it is important to mention the regulation of the key precursor of NAD^+^ synthesis, nicotinamide. *Nicotinamide N-methyltransferase* (NNMT) reduces the nicotinamide level inside the cell available for the energy metabolism by methylation, but at the same time hypomethylation of histones and other cancer-related proteins combined with heightened expression of pro-tumorigenic gene products occurs. Increased NNMT enzyme activity may determine a decrease in availability of nicotinamide. Overexpression of NNMT has been reported in a wide range of malignancies, including melanoma [33,34]. Walczak et al. studied the effect of UVB radiation on cell death in melanoma SK-MEL-3 cells exposed to Kyn and KYNA. The value of necrotic cells increased in the presence of Kyn, but not in the presence of KYNA at higher concentrations (over 10^−3^ mM) after UVB exposure [35]. Melatonin, intermediate of the serotonin pathway, is a key component in the body’s defense in many pathological conditions. It is a powerful antioxidant protecting against aging, ischemic and reperfusion injury, acts as immunomodulator, suppressor of inflammation and mitochondrial dysfunction, counteracts the Warburg effect, inhibits proliferation, and promotes apoptosis in various cancer models [36]. The synthesis of melatonin decreases with age and in some pathological conditions, with the largest decline observed in Alzheimer’s disease, cardiovascular issues, and cancer. Due to the activation of Trp metabolism by inflammation and stress via the Kyn pathway, Trp is depleted and the metabolization of Trp in the direction of melatonin production is suppressed, which has unfortunate consequences, especially for critically ill patients [37]. A significantly lower ratio of the final metabolite of the serotonin pathway, 5-HIAA to Trp was observed in the present study in melanoma patients. From this fact, we concluded that inhibition of the metabolic serotonin pathway occurred in our melanoma patients. On the other hand, there is an indication that the melanin metabolite, DHICA, may also be produced via the noncanonical precursor, a serotonin pathway intermediate, via 5-HT [38]. The concentration of DHICA as well as the DHICA/Trp ratio was measured at significantly higher levels in melanoma patients compared to healthy individuals, and the highest values were recorded in patients with melanoma stage IA, while the urinary concentration of DHICA decreased with higher stage.

The microbial community is a key component of the tumor microenvironment that influences the initiation, support, and response to cancer treatment. It follows that the microbial metabolism of tryptophan–indole is altered in carcinogenesis [25]. The bacterial metabolite of Trp, indoxyl sulfate, is considered a uremic toxin because its concentration is significantly increased in chronic kidney disease and is proposed to be one of the factors linking kidney dysfunction with an increased risk of developing cardiovascular disease. Sári et al. pointed to the cytostatic properties of indoxyl sulfate, a bacterial metabolite of Trp, in breast cancer [39]. Recent studies have shown increased urinary IS concentrations in melanoma patients [4,5]. In our previous study, we pointed to a correlation between DHICA, Trp, and IS urinary excretions in melanoma patients. These metabolites belong to the tryptophan–indole metabolic pathway and all act in melanogenesis but without unknown links to their mutual metabolic pathways [4].

On the other hand, tryptophan bacterial catabolism is suppressed in breast cancer, and a higher expression of indoxyl sulfate producing liver enzymes leads to better survival, suggesting that the cytostatic properties conferred by indoxyl sulfate production are suppressed in breast cancer. Konopelski and Mogilnicka investigated the biological actions of indole-3-propionic acid (IPA), intermediate of the indole pyruvate pathway, to prove that gut bacteria-derived metabolites of tryptophan share the biological effects of their precursor [40]. IPA has a positive role at the cellular level by preventing oxidative stress damage, lipoperoxidation and inhibiting pro-inflammatory cytokine synthesis. Vyhlídalová et al. characterized the interactions of intestinal microbial catabolites of tryptophan with the aryl hydrocarbon receptor (AhR). Several compounds formed in vivo and in vitro from Trp act as ligands and activators of AhR. These include Trp photoreactive catabolites formed in the skin after UV irradiation, endogenous Trp metabolites or microbial metabolites produced in the skin or intestines [24].

## 4. Materials and Methods

### 4.1. Composition of the Study Group

The study group consisted of 82 patients with pigmented malignant melanoma (clinical stage 0–IV; average age 57 ± 15 years; men and women ratio 44:38), and 51 healthy controls (average age 37 ± 11 years; men and women ratio 35:16) in a total volume of 133 people, with an average age of 49 ± 17 years, with a minimum of 17 years and a maximum of 85 years. Categorization of the volunteers was based on this selection with respect to melanoma clinical stages. Assigned numbers of the categories were as follows: healthy control group; 0—malignant melanoma patients clinical stage 0; 1—malignant melanoma patients clinical stage IA; 2—malignant melanoma patients clinical stage IB and IIA; 3—malignant melanoma patients clinical stage IIB, IIC, and IIIA; 4—malignant melanoma patients clinical stage IIIB and IIIC; 5—malignant melanoma patients clinical stage IV.

Patients were recruited during hospitalization at the Department of Plastic and Reconstructive Surgery UPJŠ LF in Košice.

The diagnosis of malignant melanoma was confirmed histologically by the anatomic stage grouping for cutaneous melanoma using clinical staging (0–IV) based on the Breslow thickness of the melanoma, the degree of spread to regional lymph nodes, and by the presence of distant metastasis.

The healthy control group was chosen by random assignment, with the following criteria: absence of any disease, negative hematological and biochemical laboratory tests, negative family anamnesis.

Written informed consent was obtained from all patients prior to sample collection. All clinical investigations were conducted in accordance with the Declaration of Helsinki, and the study was approved by the Ethics Committee of the University of P.J. Šafárik in Košice, Medical Faculty (20N/2016).

### 4.2. Chemicals and Reagents

#### 4.2.1. Chemicals and Reagents

Acetonitrile (ACN), creatinine, IS, KYNA, Kyn, Trp, and 5-HIAA were purchased from Sigma–Aldrich (St. Louis, MO, USA), DHICA from Toronto Research Chemicals Inc. (Toronto, ON, Canada), formic acid from Riedel-de Haën (Seelze, Germany). Ultrapure water was prepared by ultrafiltration of distilled water using the Simplicity system (Millipore, Molsheim, France). Mobile phase of 15% acetonitrile (ACN:H_2_O; 15:85) was prepared in deionized water with an addition of 0.05% formic acid. Stock solutions of creatinine, DHICA, IS, KYNA, Kyn, Trp, and 5-HIAA were prepared by diluting given compounds to a concentration of 1 mg/mL in mobile phase. All reagents were of analytical grade.

#### 4.2.2. Preparation of Standard Samples and Samples of the Experimental Study Group

Stock solutions of studied metabolites were diluted to obtain standard mixtures in the range of physiological and pathological values in human urine as follows: creatinine: 14–24–84–164 ppm; DHICA: 0.1–0.2–0.3 ppm; IS: 2–10–20 ppm; KYNA: 5–25–50 ppm; Kyn: 1–5–10 ppm; Trp: 1–5–10 ppm; 5-HIAA: 1–3–5 ppm.

Urine samples for RP-HPLC analysis were obtained from patients with malignant melanoma immediately following admission to the hospital. Participants of healthy controls were given a morning appointment and asked to fast at least 8 h before the sample collection.

Urine samples were taken under standard conditions as first morning urine. Samples were stored at −27 °C. After thawing and centrifugation at 5000 rpm for 10 min at laboratory temperature (Centrifuge Boeco S8, Hamburg, Germany), samples were filtered by PVDF syringe filters with pore size of 0.2 μm and diluted with mobile phase. For RP-HPLC analysis, 5% urine was used.

#### 4.2.3. Instrumentation

Reversed-Phase High-Performance Liquid Chromatography (RP-HPLC)

HPLC was performed in a modular reversed-phase high-performance liquid chromatography system (RP–HPLC, Schimadzu, Japan) with the use of Nucleosil expert column (EC) 100-5 C18 (Macherey–Nagel, Düren, Germany; column length 250 mm, inner diameter 4 mm, particle size 5 µm, pore size 100 Å). Samples were injected in a volume of 40 µL under isocratic conditions: mobile phase: 15% acetonitrile; flow rate: 0.8 mL/min; column temperature: 30 °C, analysis time: 30 min. Metabolites were detected by an ultraviolet–visible (UV–VIS) detector at 280 and 220 nm and by a fluorescence detector with a xenon lamp at excitation/emission wavelength 280/350 nm and 315/425 nm. The performance of the method was evaluated in terms of accuracy, linearity, imprecision, and limit of detection.

#### 4.2.4. Method Validation

Data were analyzed using STATISTICA 10 data analysis software (StatSoft Inc., Tulsa, OK, USA). Statistical analysis was performed via the following statistical tests: the Kolmogorov–Smirnov test and the Skewness–kurtosis (Jarque–Bera) test were used to determine whether a sample was normally distributed. Continuous outcome variables exhibiting a skewed distribution were analyzed by the nonparametric Mann–Whitney U test (Wilcoxon rank-sum test) of statistical significance. 

Pearson’s correlation test and Spearman’s rank order correlation test were used to determine the statistical dependence between parameters with normal and log-normal distribution, respectively.

The ranges of urine metabolites were calculated according to their log-normal distribution.

#### 4.2.5. Linearity of RP-HPLC Method

The linearity of the RP-HPLC method was established by injecting standard mixtures of the metabolites in the range of physiological and pathological values in human urine in the concentration range 14–164 ppm for creatinine; 0.1–0.3 ppm for DHICA; 2–20 ppm for IS; 5–50 ppm for KYNA; 1–10 ppm for Kyn; 1–10 ppm for Trp; and 1–5 for 5-HIAA. Representative regression equations for the calibration curves (*n* = 4) were y = 9.738 × 10^−6^x (R^2^ = 0.9999), y = 4.481 × 10^−8^x (R^2^ = 1.0000), y = 5.384 × 10^−6^x (R^2^ = 0.9998), y = 2.800 × 10^−4^x (R^2^ = 0.9804), y = 4.925 × 10^−6^x (R^2^ = 0.9904), y = 7.015 × 10^−8^x (R^2^ = 1.0000), and y = 1.984 × 10^−7^x (R^2^ = 0.9993) for creatinine, DHICA, IS, KYNA, Kyn, Trp, and 5-HIAA, respectively.

#### 4.2.6. Imprecision and Accuracy of RP-HPLC Method

Interday method imprecision and accuracy were determined by replicate analysis (*n* = 2) of the same five urine samples in which the values of DHICA, KYNA, Kyn, Trp, 5-HIAA, and IS were calculated on 10 consecutive days. The instrument imprecision was determined by repetitive injections of urine samples containing the same concentration of IS (0.01 mg/L) from the same vial performed on the same day.

#### 4.2.7. Limit of Detection of RP-HPLC Method

The limit of detection was determined by serial dilutions of working solutions to obtain a signal/noise ratio of approximately 3:1.

## 5. Conclusions

Malignant melanoma is one of the most aggressive forms of skin cancer due to its metastatic potential. The diagnostic features of melanoma range from subtle to overt characteristics. A well-established guide to the examination and interpretation of pigmented lesions has been the ABCDE acronym, which stands for asymmetry, border irregularity, color variations, diameter, and evolving. A higher number of ABCDE signs is associated with a higher probability of recognizing malignant melanoma, even by a nondermatologist, but not in situ diagnosis of melanoma. Reliance on the ABCDE rule alone may lead to missed early identifications of malignant melanoma lesions. Unfortunately, only histology can unequivocally confirm the clinical diagnosis of melanoma, but in many cases, this only happens in the later stages of the disease. Altered metabolism is typical for various diseases. By examining and recognizing suitable markers characterizing these conditions, the onset of a specific change in metabolism can be predicted. One such pathway is tryptophan metabolism, the processing of which can be influenced by both endogenous and exogenous factors. Its metabolites are demonstrably affected in malignant melanoma. DHICA, which is a common metabolite of both the tryptophan and tyrosine pathways, has the highest concentration in stage IA patients in our cohort, while its concentration decreases in higher stages. Further research is needed to verify the hypothesis whether DHICA is a suitable marker for the early detection of malignant melanoma, which would contribute to the prevention as well as to the initiation of successful treatment of malignant melanoma.

## Figures and Tables

**Figure 1 ijms-23-09160-f001:**
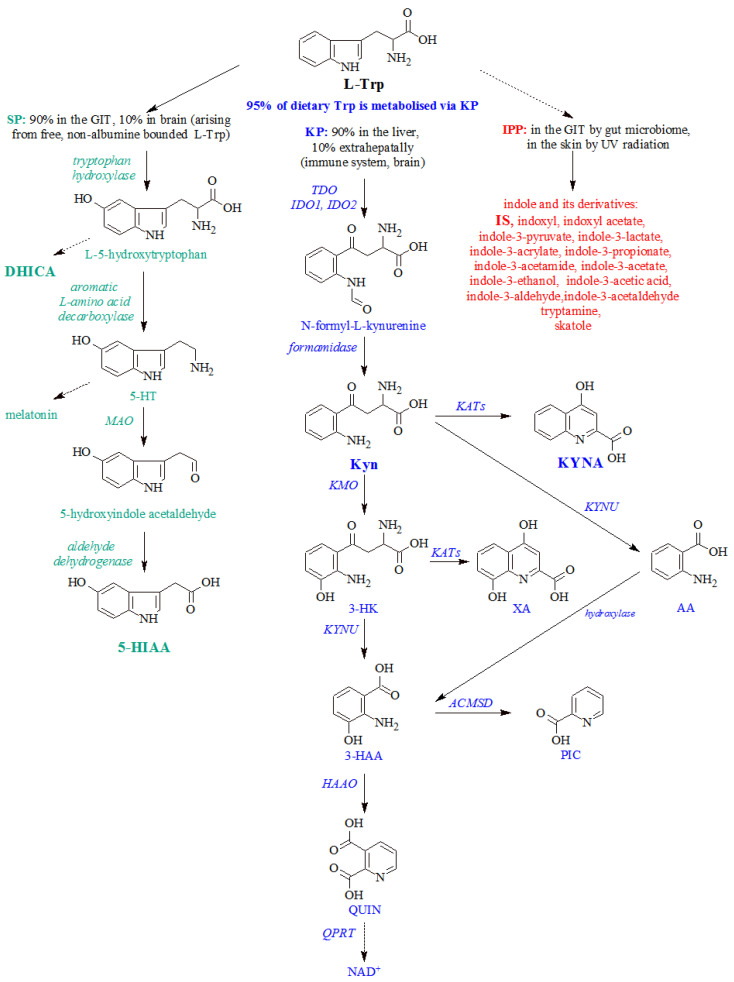
The three pathways of tryptophan metabolization are the kynurenine pathway, serotonin pathway, and the indole pathway. KP—kynurenine pathway, SP—serotonin pathway, IPP—indole pyruvate pathway; L-Trp—L-tryptophan, Kyn—kynurenine, 3-HK—3-hydroxykynurenine, 3-HAA—3-hydroxyanthranilic acid, QUIN—quinolinic acid, NAD^+^—nicotinamide adenine dinucleotide, KYNA—kynurenic acid, AA—anthranilic acid, XA—xanthurenic acid, PIC—picolinic acid, 5-HT—serotonin, 5-HIAA—5-hydroxyindole-3-acetic acid, DHICA—5,6-dihydroxyindole-2-carboxylic acid, IS—indoxyl sulfate; TDO—*tryptophan 2,3-dioxygenase*, IDO1—*indoleamine 2,3-dioxygenase 1*, IDO2—*indoleamine 2,3-dioxygenase 2*, KMO—*kynurenine 3-monooxygenase*, KYNU—*kynureninase*, HAAO—*3-hydroxyanthranilic acid oxygenase*, QPRT—*quinolinic acid phosphoribosyltransferase*, KATs—*kynurenine aminotransferases*, ACMSD—*2-amino-3-carboxymuconate-6-semialdehyde decarboxylase*, MAO—*monoaminooxygenase*; dashed arrows include more catalytic reaction steps; metabolites analyzed in this study are highlighted in bold.

**Figure 2 ijms-23-09160-f002:**
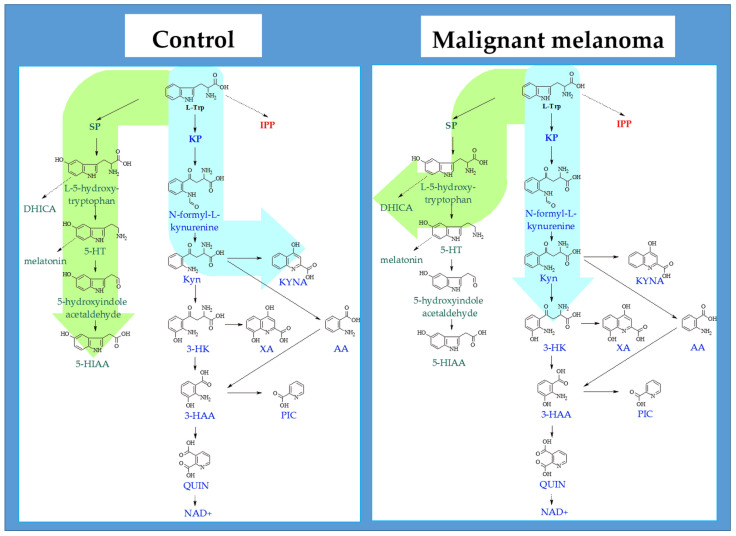
Comparison of the metabolization of the selected intermediates of the kynurenine, serotonin, and the indole pathway in malignant melanoma and in healthy control group. KP—kynurenine pathway, SP—serotonin pathway, IPP—indole pyruvate pathway; L-Trp—L-tryptophan, Kyn—kynurenine—intermediate in NAD^+^ biosynthesis but also neurotoxic intermediates 3-HK and QUIN, 3-HK—3-hydroxykynurenine, 3-HAA—3-hydroxyanthranilic acid, QUIN—quinolinic acid, NAD^+^—nicotinamide adenine dinucleotide, KYNA—kynurenic acid—product of KP branch, neuroprotective compound, AA—anthranilic acid, XA—xanthurenic acid, PIC—picolinic acid, 5-HT—serotonin, 5-HIAA—5-hydroxyindole-3-acetic acid—final product of SP, DHICA—5,6-dihydroxyindole-2-carboxylic acid—intermediate in melanin biosynthesis, IS—indoxyl sulfate—product of microbial degradation, uremic toxin, cardiotoxin.

**Figure 3 ijms-23-09160-f003:**
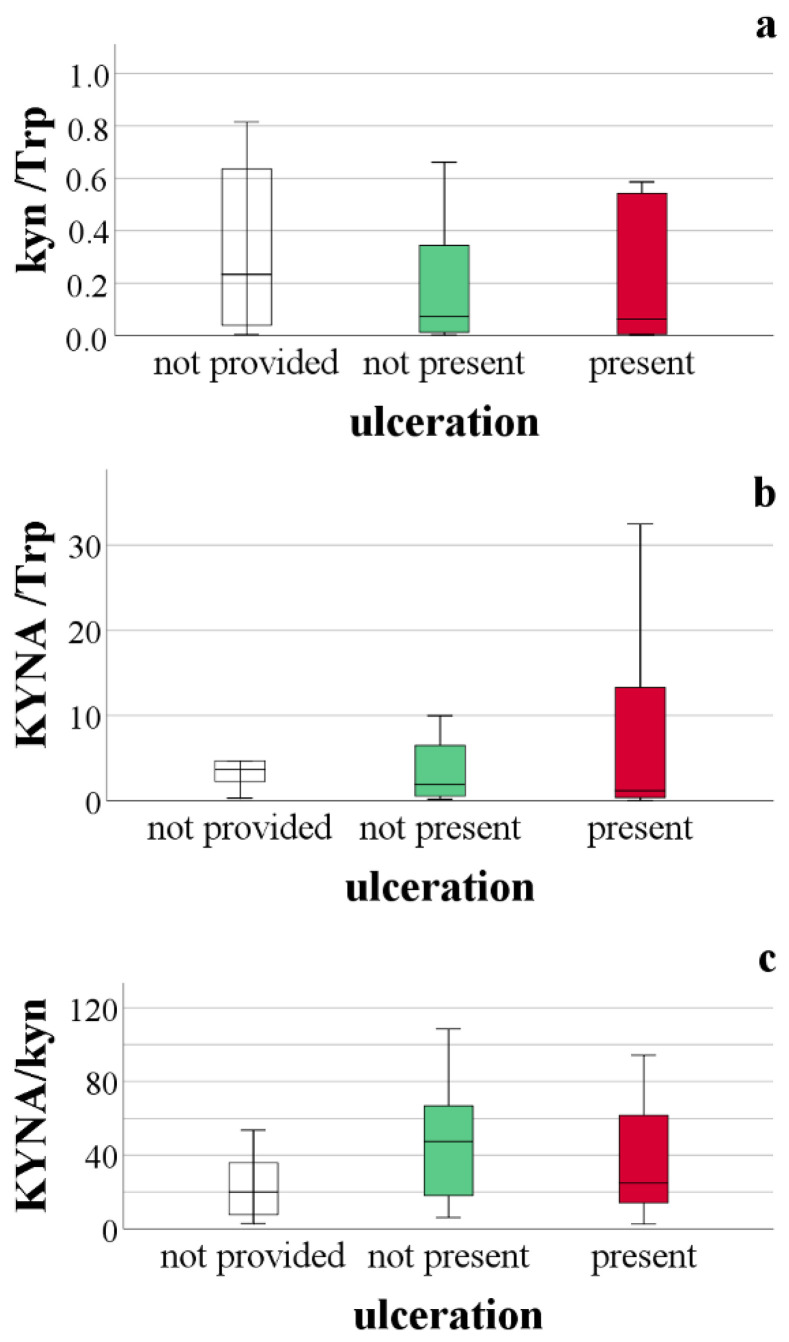
Comparison of kynurenine pathway metabolites based on the presence of ulceration. (**a**) comparison of kynurenine level relative to tryptophan concentration, based on the presence of ulceration, *p* = 0.3199; (**b**) comparison of kynurenic acid level relative to tryptophan concentration, based on the presence of ulceration, *p* = 0.412; (**c**) comparison of kynurenic acid level relative to kynurenine concentration, based on the presence of ulceration, *p* = 0.209.

**Figure 4 ijms-23-09160-f004:**
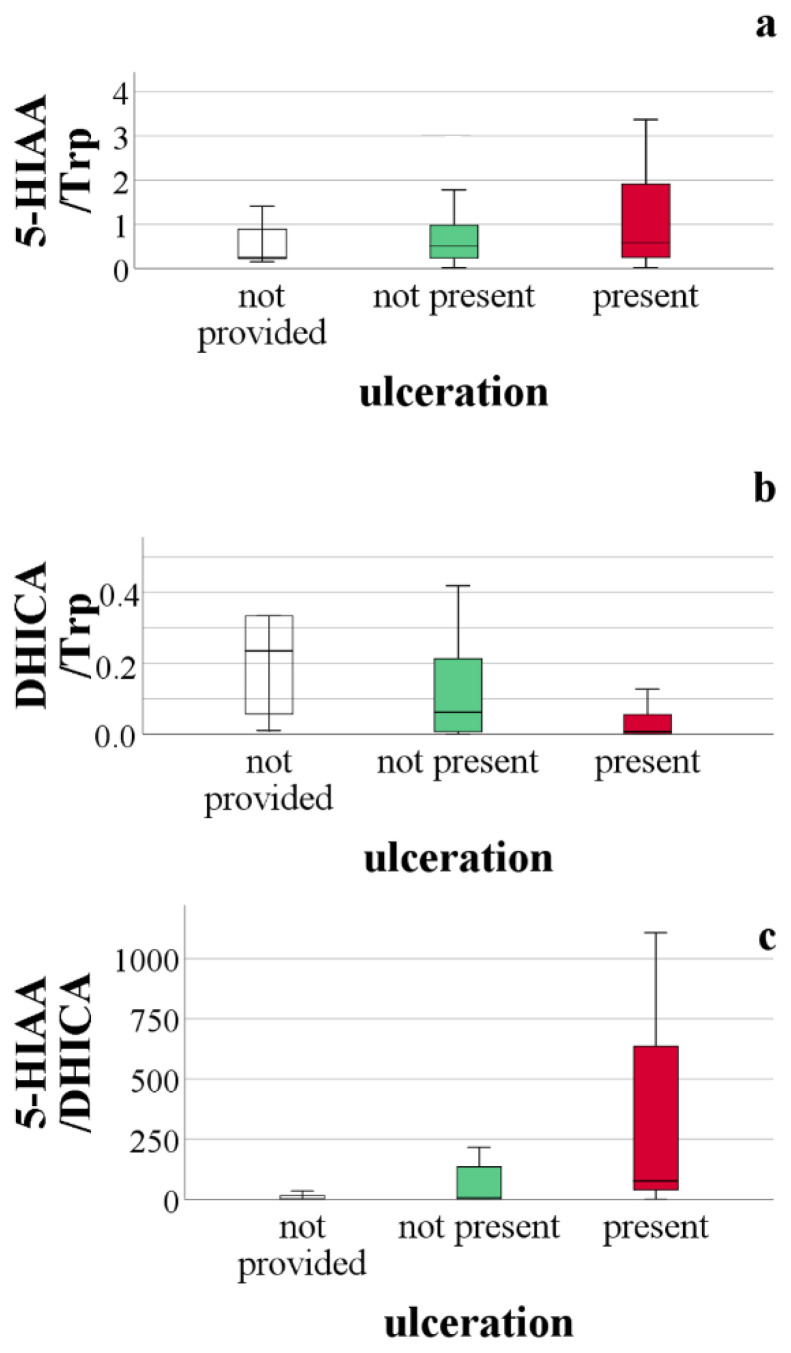
Comparison of serotonin pathway metabolites based on the presence of ulceration. (**a**) comparison of 5-hydroxyindole-3-acetic acid level relative to tryptophan concentration, based on the presence of ulceration, *p* = 0.223; (**b**) comparison of 5,6-dihydroxyindole-2-carboxylic acid level relative to tryptophan concentration, based on the presence of ulceration, *p* = 0.171; (**c**) comparison of 5-hydroxyindole-3-acetic acid level relative to 5,6-dihydroxyindole-2-carboxylic acid concentration, based on the presence of ulceration, *p* = 0.210.

**Figure 5 ijms-23-09160-f005:**
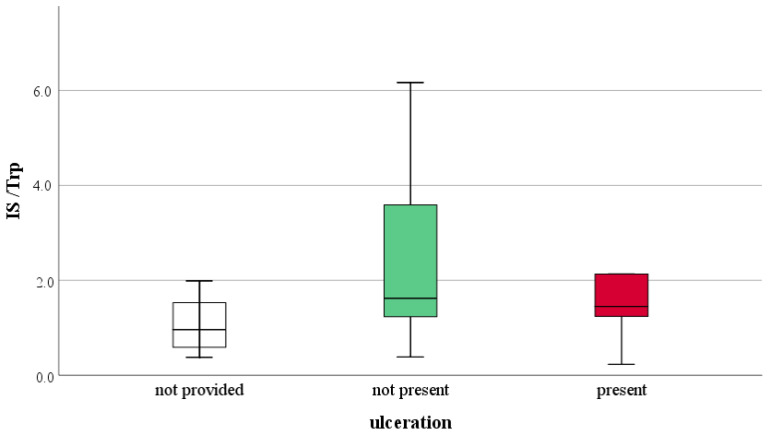
Comparison of indole pyruvate pathway metabolite, the indoxyl sulfate level relative to tryptophan concentration, based on the presence of ulceration*, p* = 0.493.

**Figure 6 ijms-23-09160-f006:**
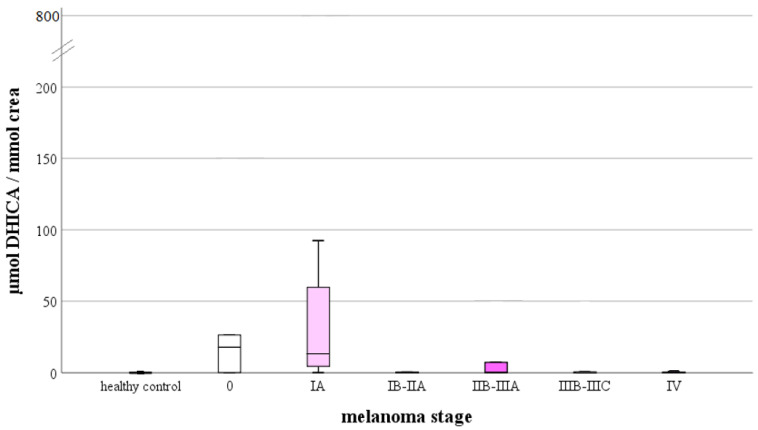
Comparison of 5,6-dihydroxyindole-2-carboxylic acid level relative to urinary creatinine concentration, based on the melanoma stages.

**Table 1 ijms-23-09160-t001:** Composition of respondents by gender in individual categories based on melanoma stage.

Melanoma Stage	Women		Men	
	Frequency	Percent	Frequency	Percent
**Healthy control**	16	29.6	35	44.3
**Stage 0**	4	7.4	2	2.5
**Stage IA**	10	18.5	14	17.7
**Stage IB**	1	1.9	3	3.8
**Stage IIA**	4	7.4	3	3.8
**Stage IIB**	6	11.1	2	2.5
**Stage IIC**	0	0.0	1	1.3
**Stage IIIA**	0	0.0	2	2.5
**Stage IIIB**	7	13.0	7	8.9
**Stage IIIC**	2	3.7	5	6.3
**Stage IV**	4	7.4	5	6.3
**total**	54	100	79	100

The background color indicates the severity of the melanoma stage; the darker the shade, the greater the severity.

**Table 2 ijms-23-09160-t002:** Urinary Trp, Kyn, KYNA, 5-HIAA, DHICA, and IS levels.

	Melanoma Patients, *n* = 82		Healthy Control, *n* = 51		Mann–Whitney U Test *p*-Value
	Average ± st.dev.	Median; IQR	Average ± st.dev.	Median; IQR	
**Trp [µmol/mmol]**	88.03 ± 121.02	33.47; 125.47	4.29 ± 3.03	3.46; 6.3	1.09 × 10^−14^ ***
**Kyn [µmol/mmol]**	51.17 ± 66.13	25.19; 73.12	10.02 ± 14.73	2.34; 14.70	3.94 × 10^−10^ ***
**KYNA [µmol/mmol]**	1363.47 ± 1978.01	699.39; 1100.92	253.67 ± 211.97	202.01; 191.84	3.99 × 10^−14^ ***
**5-HIAA ^b^ [µmol/mmol]**	24.84 ± 28.23	11.56; 30.87	2.94 ± 1.94	2.62; 1.89	2.53 × 10^−15^ ***
**DHICA ^a^ [µmol/mmol]**	29.61 ± 100.37	0.44; 15.91	0.01 ± 0.02	0.00; 0.01	8.13 × 10^−16^ ***
**IS [µmol/mmol]**	136.16 ± 185.80	76.31; 154.27	6.18 ± 5.04	5.00; 6.58	5.88 × 10^−18^ ***

^a^ *n* = 72 melanoma patients, ^b^ *n* = 74 melanoma patients; Trp—tryptophan; Kyn—kynurenine; KYNA—kynurenic acid; DHICA—5,6-dihydroxyindole-2-carboxylic acid; IS—indoxyl sulfate; 5-HIAA—5-hydroxyindole-3-acetic acid; crea—creatinine; ***—difference is significant at the 0.001 level.

**Table 3 ijms-23-09160-t003:** Urinary Kyn/Trp, KYNA/Trp, and KYNA/Kyn ratios characteristic of the kynurenine pathway.

	Melanoma Patients, *n* = 82		Healthy Control, *n* = 51		Mann–Whitney U Test *p*-Value
	Median	IQR	Median	IQR	
**Kyn/Trp**	0.06	0.37	0.01	0.16	0.004 **
**KYNA/Trp**	1.50	5.49	0.99	2.10	0.042 *
**KYNA/Kyn**	44.30	46.55	108.56	393.80	0.071 ^ns^

Kyn—kynurenine; Trp—tryptophan; KYNA—kynurenic acid; *—difference is significant at the 0.005 level; **—difference is significant at the 0.010 level; ^ns^—nonsignificant difference.

**Table 4 ijms-23-09160-t004:** Urinary 5-HIAA/Trp, DHICA/Trp, and 5-HIAA/DHICA ratios characteristic of the serotonin pathway and IS/Trp characteristic of the indole pathway.

	Melanoma Patients, *n* = 82		Healthy Control, *n* = 51		Mann-Whitney U Test *p*-Value
	Median	IQR	Median	IQR	
**5-HIAA/Trp**	0.48	0.80	0.83	1.32	0.013 *
**DHICA/Trp**	0.05	0.17	0.00	0.01	2.54 × 10^−11^ ***
**5-HIAA/DHICA**	21.68	159.73	0.00	185.93	0.003 **
**IS/Trp**	1.46	1.01	1.63	4.10	0.588 ^ns^

5-HIAA—5-hydroxyindole-3-acetic acid; Trp—tryptophan; DHICA—5,6-dihydroxyindole-2-carboxylic acid; IS—indoxyl sulfate; *—difference is significant at the 0.005 level; **—difference is significant at the 0.010 level; ***—difference is significant at the 0.001 level; ^ns^—nonsignificant difference.

**Table 5 ijms-23-09160-t005:** Nonparametric correlation between the studied parameters, the Breslow thickness, Clark level, and the melanoma stage.

		Trp	Kyn	KYNA	5-HIAA	DHICA	IS
**Kyn**	rho	0.474 **					
	*p*-value	8.17 × 10^−9^					
**KYNA**	rho	0.728 **	0.541 **				
	*p*-value	3.36 × 10^−23^	1.73 × 10^−11^				
**5-HIAA**	rho	0.688 **	0.578 **	0.69 1**			
	*p*-value	7.94 × 10^−19^	1.76 × 10^−12^	4.69 × 10^−19^			
**DHICA**	rho	0.689 **	0.470 **	0.702 **	0.646 **		
	*p*-value	1.35 × 10^−18^	4.10 × 10^−8^	1.50 × 10^−19^	1.20 × 10^−15^		
**IS**	rho	0.741 **	0.569 **	0.809 **	0.675 **	0.783 **	
	*p*-value	1.97 × 10^−24^	9.36 × 10^−13^	4.83 × 10^−32^	5.79 × 10^−18^	1.03 × 10^−26^	
**Breslow thickness**	rho	−0.130	0.028	−0.064	−0.246	−0.361 **	−0.136
	*p*-value	0.285	0.821	0.601	0.054	0.005	0.261
**Clark level**	rho	0.069	0.143	0.079	−0.139	−0.184	−0.007
	*p*-value	0.568	0.235	0.513	0.276	0.155	0.953
**Melanoma stage**	rho	−0.151	0.030	−0.165	−0.320 **	−0.491 **	−0.229 *
	*p*-value	0.177	0.789	0.137	0.005	1.19 × 10^−5^	0.038

Trp—tryptophan in µmol/mmol of creatinine; Kyn—kynurenine in µmol/mmol of creatinine; KYNA—kynurenic acid in µmol/mmol of creatinine; DHICA—5,6-dihydroxyindole-2-carboxylic acid in µmol/mmol of creatinine; IS—indoxyl sulfate in µmol/mmol of creatinine; 5-HIAA—5-hydroxyindole-3-acetic acid in µmol/mmol of creatinine; rho—Spearman’s rho correlation coefficient; significance 2-tailed; *—correlation is significant at the 0.005 level; **—correlation is significant at the 0.010 level.

**Table 6 ijms-23-09160-t006:** Nonparametric correlation between the ratios of the studied parameters, the Breslow thickness, Clark level, and the melanoma stage.

		Kyn/Trp	KYNA/Trp	KYNA/Kyn	5-HIAA/Trp	DHICA/Trp	5-HIAA/DHICA	IS/Trp
**KYNA/Trp**	rho	0.807 **						
	*p*-value	1.07 × 10^−31^						
**KYNA/Kyn**	rho	−0.665 **	−0.176 *					
	*p*-value	2.66 × 10^−18^	0.043					
**5-HIAA/Trp**	rho	0.209 *	0.166	−0.194 *				
	*p*-value	0.019	0.065	0.030				
**DHICA/Trp**	rho	0.289 **	0.264 **	−0.086	−0.093			
	*p*-value	0.001	0.003	0.345	0.308			
**5-HIAA/DHICA**	rho	0.187 *	0.139	−0.175	0.193 *	0.131		
	*p*-value	0.040	0.128	0.055	0.034	0.153		
**IS/Trp**	rho	0.179 *	0.160	−0.101	0.521 **	0.223 *	0.065	
	*p*-value	0.039	0.067	0.249	4.86 × 10^−10^	0.013	0.481	
**Breslow thickness**	rho	0.011	0.014	−0.064	0.039	−0.326 *	0.296 *	−0.019
	*p*-value	0.926	0.911	0.601	0.766	0.011	0.024	0.875
**Clark level**	rho	0.034	0.056	−0.136	−0.209	−0.244	0.104	−0.126
	*p*-value	0.777	0.643	0.257	0.101	0.058	0.434	0.294
**Melanoma stage**	rho	0.032	−0.009	−0.159	−0.012	−0.516 **	0.394 **	−0.092
	*p*-value	0.775	0.935	0.154	0.921	3.48 × 10^−6^	0.001	0.409

Trp—tryptophan in µmol/mmol of creatinine; Kyn—kynurenine in µmol/mmol of creatinine; KYNA—kynurenic acid in µmol/mmol of creatinine; DHICA—5,6-dihydroxyindole-2-carboxylic acid in µmol/mmol of creatinine; IS—indoxyl sulfate in µmol/mmol of creatinine; 5-HIAA—5-hydroxyindole-3-acetic acid in µmol/mmol of creatinine; rho—Spearman’s rho correlation coefficient; significance 2-tailed; *—correlation is significant at the 0.005 level; **—correlation is significant at the 0.010 level.

## Data Availability

The database of aggregated statistics prepared for analysis is stored in secure, confidential, password protected storage in the server of the Department of Medical and Clinical Biochemistry, Pavol Jozef Šafárik University in Košice, Faculty of Medicine. The data has been anonymized. Completely deidentified records could be made available to interested persons/organizations on request to the corresponding author at beata.hubkova@upjs.sk.

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
