# Peer review of "Tryptophan: Its Metabolism along the Kynurenine, Serotonin, and Indole Pathway in Malignant Melanoma"

_ijms, 2022, doi:10.3390/ijms23169160_

Round 1

Reviewer 1 Report

The manuscript “Tryptophan: Its metabolism along the kynurenine, serotonin and indole pathway in malignant melanoma” is a research article that attempted to identify altered metabolism of tryptophan pathways in melanoma patients. The manuscript is generally well-written, few typos are present. However, there are some concerns that authors must address in order to consider the manuscript for publication:

1.       The main concern of this experimental work is that an important part of the tryptophan/kynurenine pathway is missing. Indeed, the authors completely ignored nicotinamide and its main regulator, the enzyme nicotinamide N-methyltransferase. Nicotinamide can be introduced by diet or can be synthesized starting from the tryptophan. Nicotinamide can be converted into NAD in the NAD salvage pathway, while the methylation of the nicotinamide by nicotinamide N-methyltransferase can impact the intracellular NAD content and thus the activities of NAD-dependent enzymes like PARPs and sirtuins. All these factors have been demonstrated to be impaired in cancer cell metabolism. An overexpression of NNMT has been reported in several malignancies, including melanoma (PMID: 34638427; PMID: 29420365). Due to the overexpression of the NNMT in melanoma, the reduced content of tryptophan measured in serum could be the consequence of an enhanced nicotinamide synthesis starting from tryptophan. Authors should incorporate these observations in the discussion section, since this reinforces the reliability of data obtained in this study.

2.       Figure 1: the quality of the figure is very poor. Please improve it. Moreover, the legend is written as part of a paragraph. Please fix it.

3.       Figure 2: lines 224-232 should be part of the figure legend.

4.       Table 2: values from melanoma patients and controls should be separated in a clearer way, since the table looks confusing.

Author Response

Dear reviewer.

Thank you for taking the time to improve our article. Your questions are an inspiration for our further work in this area. Please find below our answers to your questions and suggestions:

The manuscript “Tryptophan: Its metabolism along the kynurenine, serotonin and indole pathway in malignant melanoma“ is a research article that attempted to identify altered metabolism of tryptophan pathways in melanoma patients. The manuscript is generally well-written, few typos are present.

A: The manuscript was checked, and typos were corrected.

However, there are some concerns that authors must address in order to consider the manuscript for publication: The main concern of this experimental work is that an important part of the tryptophan/kynurenine pathway is missing. Indeed, the authors completely ignored nicotinamide and its main regulator, the enzyme nicotinamide N-methyltransferase. Nicotinamide can be introduced by diet or can be synthesized starting from the tryptophan. Nicotinamide can be converted into NAD in the NAD salvage pathway, while the methylation of the nicotinamide by nicotinamide N-methyltransferase can impact the intracellular NAD content and thus the activities of NAD-dependent enzymes like PARPs and sirtuins. All these factors have been demonstrated to be impaired in cancer cell metabolism. An overexpression of NNMT has been reported in several malignancies, including melanoma (PMID: 34638427; PMID: 29420365). Due to the overexpression of the NNMT in melanoma, the reduced content of tryptophan measured in serum could be the consequence of an enhanced nicotinamide synthesis starting from tryptophan. Authors should incorporate these observations in the discussion section, since this reinforces the reliability of data obtained in this study.

A: We greatly appreciate your comment regarding discussing the topic of nicotinamide and its regulator in view of its prognostic significance in melanoma. Based on your comment, we have included the following sentences with relevant references in the text of the discussion:

“In this regard, it is important to mention the regulation of the key precursor of NAD+ synthesis, nicotinamide. Nicotinamide N-methyltransferase (NNMT) reduces the nicotinamide level inside the cell available for energy metabolism by methylation, but at the same time hypomethylation of histones and other cancer-related proteins combined with heightened expression of protumorigenic gene products occurs. Increased NNMT enzyme activity may determine a decrease in the availability of nicotinamide. Overexpression of NNMT has been reported in a wide range of malignancies, including melanoma (33, 34)“.

  1. Figure 1: the quality of the figure is very poor. Please improve it. Moreover, the legend is written as part of a paragraph. Please fix it.

A: The quality of Figure 1 has been improved and the legend has been added to the Figure title.

  1. Figure 2: lines 224-232 should be part of the figure legend.

A: The legend has been added to the Figure title.

  1. Table 2: values from melanoma patients and controls should be separated in a clearer way, since the table looks confusing.

A: The values of melanoma patients and controls have been formatted to achieve a more readable and clearer table.

Reviewer 2 Report

A review of the manuscript titled „Tryptophan: Its metabolism along the kynurenine, serotonin and indole pathway in malignant melanoma” by Beáta Hubková, Marcela Valko-Rokytovská, Beáta ÄŒižmárová, Marianna Zábavníková, Mária Mareková and Anna Birková to Int. J. Mol. Sci.

In the manuscript, the authors report on tryptophan metabolism in the urine of melanoma patients. The paper is interesting. Its undoubted advantage is the use of urine, as a material that is easy to obtain.

The work is well thought out, each study is well reasoned.

Please reconsider the following:

1.       How do the authors argue that serum tryptophan levels are low in people with melanoma?

2.       The authors demonstrate that in melanoma patients, tryptophan metabolism via the kynurenine pathway is increased toward KYNA. However, it is difficult to comment on the direction of changes since only the concentration of Trp-Kyn-KYNA was determined. It is a great pity that the concentration of cytotoxic metabolites, such as 3-HK or 3-HAA, was not determined.

3.       Authors should standardize the HPLC or RP-HPLC notation throughout the manuscript.

4.       How was the creatinine concentration in urine determined?

Author Response

Dear reviewer.

Thank you for taking the time to improve our article. Your questions are an inspiration for our further work in this area. Please find below our answers to your questions and suggestions:

1. How do the authors argue that serum tryptophan levels are low in people with melanoma?

A: Thank you for pointing out the missing reference. After the following sentence: " Melanoma patients in our study showed significantly higher levels of tryptophan in the urine, which may not be inconsistent with the low serum tryptophan concentration in malignant melanoma patients with poor prognosis." we added the explanation as well as the relevant reference in the following wording: “The low level of tryptophan in the serum of patients with melanoma is explained precisely by its accelerated metabolism and degradation, which results in an increased concentration of degradation products both in serum and in urine as well as the appearance of tryptophan itself in the urine [29]. This finding is in accordance with the study of Costa et al. [30]."

 2. The authors demonstrate that in melanoma patients, tryptophan metabolism via the kynurenine pathway is increased toward KYNA. However, it is difficult to comment on the direction of changes since only the concentration of Trp-Kyn-KYNA was determined. It is a great pity that the concentration of cytotoxic metabolites, such as 3-HK or 3-HAA, was not determined.

A: In the future, we plan to determine other important metabolites of the Trp pathway to have a clearer view of the changes in metabolism in melanoma. In this work, we focused on having a Trp metabolite representative for each of the mentioned pathways. Thank you for the notice.

3. Authors should standardize the HPLC or RP-HPLC notation throughout the manuscript.

A: Thank you for your notice, we have standardized the abbreviation RP-HPLC throughout the text

4. How was the creatinine concentration in urine determined?

A: The urinary creatinine concentration level was determined by RP-HPLC using the standard addition calibration method as described in the Methods section: the standard creatinine solution was prepared in the range of its physiological and pathological values in human urine: 14–24–84–164 ppm.